# Genome-Wide Characterization of Salt-Responsive miRNAs, circRNAs and Associated ceRNA Networks in Tomatoes

**DOI:** 10.3390/ijms222212238

**Published:** 2021-11-12

**Authors:** Zhongyu Wang, Ning Li, Qinghui Yu, Huan Wang

**Affiliations:** 1Biotechnology Research Institute, Chinese Academy of Agricultural Sciences, Beijing 100081, China; wzy_caas@163.com; 2Institute of Horticulture Crops, Xinjiang Academy of Agricultural Sciences, Urumqi 830091, China; lining@xaas.ac.cn; 3Key Laboratory of Horticulture Crop Genomics and Genetic Improvement in Xinjiang, Urumqi 830091, China; 4College of Horticulture, Xinjiang Agricultural University, Urumqi 830052, China

**Keywords:** salt stress, tomato, miRNA, circRNA, ceRNA

## Abstract

Soil salinization is a major environmental stress that causes crop yield reductions worldwide. Therefore, the cultivation of salt-tolerant crops is an effective way to sustain crop yield. Tomatoes are one of the vegetable crops that are moderately sensitive to salt stress. Global market demand for tomatoes is huge and growing. In recent years, the mechanisms of salt tolerance in tomatoes have been extensively investigated; however, the molecular mechanism through which non-coding RNAs (ncRNAs) respond to salt stress is not well understood. In this study, we utilized small RNA sequencing and whole transcriptome sequencing technology to identify salt-responsive microRNAs (miRNAs), messenger RNAs (mRNAs), and circular RNAs (circRNAs) in roots of M82 cultivated tomato and *Solanum pennellii* (*S. pennellii*) wild tomato under salt stress. Based on the theory of competitive endogenous RNA (ceRNA), we also established several salt-responsive ceRNA networks. The results showed that circRNAs could act as miRNA sponges in the regulation of target mRNAs of miRNAs, thus participating in the response to salt stress. This study provides insights into the mechanisms of salt tolerance in tomatoes and serves as an effective reference for improving the salt tolerance of salt-sensitive cultivars.

## 1. Introduction

Salt stress can affect the germination and growth of plants, which ultimately leads to the reduction of crop yield [1,2,3,4,5]. Therefore, research on salt-tolerant crops has important ecological and economic significance. With the development of DNA sequencing technology, our understanding of plant responses to salinity stress has greatly improved. In the course of evolution, many plants have evolved various salt-tolerance mechanisms to adapt to alkaline soil environments [6]. The major physiological and metabolic pathways of plants, such as photosynthesis, protein synthesis, energy, and lipid metabolism, can be affected under salt stress [7]. Furthermore, plants have also evolved a series of mechanisms to protect themselves from the effects of salt stress, such as osmotic adjustment, ion transport, and antioxidants. These mechanisms may be ubiquitous in most plants. Cultivated tomato is a moderately salt-sensitive plant [8], while some wild species have higher salt tolerance, such as *Solanum pennellii* (*S. pennellii*) [9], *S**. peruvianum* [10], *S**. Cheesmanii* [10], and *S**. chilense* [11]. In order to breed salt-tolerant crops, we need to identify the key genes that confer salt tolerance and elucidate the specific salt-tolerance mechanisms of the wild species that are more salt-tolerant. Therefore, studying the salt-tolerance mechanisms of wild tomatoes is of momentous significance.

Previous studies on phytohormone signaling and specific transcription factors (TFs) have revealed their important roles under salt-stress conditions, including in the salt overly sensitive (SOS) pathway [12], Ca^2+^-mediated signal transduction pathway [13], ABA signaling pathway [14,15], ethylene signaling pathway [16], phospholipid signaling pathway [17], mitogen-activated protein kinase (MAPK) signaling cascade [18], and so forth. ABRE binding factors (ABFs/AREBs) can regulate the expression of corresponding genes by binding to the ABA-responsive element (ABRE). Previous investigations have shown that *ABF2/AREB1*, *ABF3*, and *ABF4/AREB2* can participate in the regulation of the ABA signaling pathway under salt-stress conditions [19]. As the last step in the ethylene synthesis pathway, ACC oxidase (ACO) can convert ACC to ethylene and is involved in salt tolerance. Under salt-stress conditions, brassinosteroid (BR) can affect the growth and development of plants in many ways and improve plants’ stress tolerance [20]. As one of the key genes involved in BR synthesis, brassinosteroid-6-oxidase (*BR6OX1*) may be regulated under salt stress. The SOS signaling pathway is a kind of Ca^2+^-related pathway and plays a vital role in conferring salt tolerance in *Arabidopsis thaliana* (Arabidopsis) [21]. The important roles of *SOS1*/*2*/*3* have been studied under salt stress [22,23,24,25,26,27,28,29]. As a calcium sensor, *SOS3* can, depending on calcium, activate *SOS2* while *SOS1* may be positively regulated by *SOS2*. The resistance to salt stress of *SOS3*-overexpressing plants is similar to that of *SOS1*-overexpressing plants [22]. As the homolog of *SOS3*, SOS3-like calcium binding protein 8/calcineurin B-like 10 (*SCABP8/CBL10*) might assume a similar function in activating *SOS2* under salt stress [21]. As one of the major classes of Ca^2+^-binding proteins, CDPKs have been confirmed to be related to salt stress and the ABA signaling pathway [30,31]. However, no studies have examined the relation between *CDPK16* and salt stress. *ABF3* has been verified to activate transcription of protein phosphatase 2C (*PP2C*), and the expression levels of *HAI1*, *HAI2*, and *HAI3* were found to be significantly down-regulated in the *areb1/areb2/abf3* triple mutant [32,33]. The ability of plants to maintain Na^+^/K^+^ homeostasis is also critical to the adaption to salt stress [34,35,36,37,38]. SKOR is a K^+^ outward rectifier, and the joint activity of SKOR and *NPF7.3/NRT1.5* has been reported to be critical to the K^+^ homeostasis of the shoot in Arabidopsis [39]. Xyloglucan endotransglucosylase/hydrolase (XTH) plays an important role in cell-wall remodeling and plant growth and development [40]. In Arabidopsis, previous studies have shown that BR signaling can regulate the expression of *XTH19* and *XTH23* and promote the development of lateral roots through transcriptional factor *BES1* to adapt to salt stress [41].

In plants, there are also many non-coding RNAs (ncRNAs) that have been found to participate in the process of gene regulation in addition to mRNAs. The biological significance of ncRNAs in abiotic stress has been increasingly recognized in recent years. MicroRNAs (miRNAs) are a kind of endogenous single-stranded ncRNA found in different organisms [42]. miRNAs are about 21 to 25 nt and evolutionarily conserved. miRNAs can participate in the process of plant stress-resistance by degrading or inhibiting the expression of stress-related mRNAs through the complementary pairing principle. In recent years, researchers have found that miRNAs play an important role in the responses of plants to stresses such as salt stress [43,44,45,46,47], drought stress [48,49,50,51,52,53], and chilling stress [54,55,56,57,58]. Sunkar et al. (2004) found that miR393 could be strongly induced under salt and chilling treatment. In Arabidopsis, the expression levels of miR397b and miR402 were found to be up-regulated, and miR319c was only induced by cold treatment and it did not respond to salt stress [59]. Liu et al. (2008) demonstrated that miR165/167/168/171/319/393/396 can be induced by high salt stress in Arabidopsis [60]. miR171b/f, miR172a/b/d, and miR398 were found to be down-regulated in rice and Arabidopsis [61,62,63], while miR156a/171c/277/393a/530a/1444a were up-regulated in *Populus euphratica* under salt stress [64]. In 2011, Frazier et al. found that the expression levels of miR172/395/396 in tobacco seedlings were up-regulated under salt stress [51]. In a study from 2014, miR156b/c, miR162b, miR164c, miR166d, miR167a/b, miR171b/e, miR172a, miR319a, miR397a, miR399b, and miR5300 were found to be significantly induced in roots of *S**. linnaeanum* [65]. Moreover, miR414c was also found to affect the salt tolerance of cotton by regulating reactive oxygen species metabolism [66], and miR408 was found to remodel the ABA signaling pathway and osmoprotective biosynthesis under salt stress in wheat [67].

Circular RNAs (circRNAs) are another type of emerging endogenous ncRNA separate from miRNAs and long non-coding RNAs (lncRNAs). More and more studies have shown that circRNAs are abundant in plants and animals [68,69,70]. With the advancement of RNA research technology, researchers have found a large number of circRNAs in various organisms with important biological functions. circRNAs can act as miRNA sponges to regulate miRNA activity by binding to transcriptional regulatory elements or they can interact with proteins to regulate gene transcription and so forth [71,72,73,74]. Subsequently, circRNAs have been widely found in *Oryza sativa* L. [68], *Zea mays* [75,76], *S**. lycopersicum* [77], *Triticum aestivum* L. [78], *Glycine max* [79], and *Gossypium hirsutum* L. [80]. Recent studies have found that circRNAs can play an important role in plant development and stress resistance. Zuo et al. (2016) determined that 163 out of 854 circRNAs were regulated by chilling stress in *S**. lycopersicum* [77]. Similarly, 62 and 33 differentially expressed circRNAs (DE-circRNAs) were identified in leaves of wheat seedlings and Birch-leaf pear under dehydration stress, respectively [78,81]. Zhu et al. (2019) identified 1934 and 44 salt-responsive circRNAs in the roots and leaves of cucumber under salt stress, respectively [82]. Recently, Zhang et al. (2019) reported that host genes of circRNAs were enriched for single nucleotide polymorphisms (SNPs) associated with drought tolerance in 368 inbred maize lines, which suggests the important roles of circRNAs in maize under drought stress [83]. These observations confirm that circRNAs are involved in several biological processes, such as abiotic stress, in plants. However, the mechanisms of circRNAs under salt stress have not been studied systematically.

In 2011, the communication between ncRNAs using miRNA response elements (MREs) was first reported, with this communication constituting a competitive endogenous RNA (ceRNA) network [84]. The ceRNA hypothesis demonstrates the complex ceRNA networks formed by different types of RNAs mediated by miRNAs. It refers to mRNAs and ncRNAs, such as pseudogenes, circRNAs, and lncRNAs. Recent studies have shown that circRNAs can act as ceRNAs to regulate target RNAs. For instance, 61 circRNAs have been found to act as potential ceRNAs that can bind to miRNAs, while some miRNAs have been identified to be involved in the ethylene signaling pathway [85]. This indicates that the ethylene pathway in tomatoes might be regulated by circRNAs. Two groups discovered that endogenous circRNAs can act as miRNA sponges at the same time, which demonstrated that circRNAs play important roles as ceRNAs [71,86].

Tomato (*S**. lycopersicum*) is one of the main vegetable crops grown in China. Salt-induced damage can seriously affect the yield of tomatoes. However, research on tomato ncRNAs in response to salt stress is limited. In the present study, we identified salt-responsive miRNAs, mRNAs, and circRNAs in cultivated tomato M82 and wild tomato *S. pennellii* roots under salt stress. Based on the ceRNA hypothesis, we established the salt-responsive ceRNA (circRNA–miRNA–mRNA) regulatory networks of M82 and *S. pennellii*. These ceRNA networks can be utilized to analyze the functions of salt-responsive miRNAs and circRNAs. These results serve as a good theoretical basis for further understanding of and research on the specific salt-tolerance mechanisms of wild tomato species.

## 2. Results

### 2.1. Small RNA Sequencing of Tomato Roots

To identify salt-responsive miRNAs in M82 and *S. pennellii*, eight small RNA libraries were constructed with two replicates for the roots of cultivated tomato M82 and wild tomato *S. pennellii* with and without salt treatment. In total, 164,283,335 raw reads were generated (Table 1). After removing the low-quality and adapter sequences, 163,027,394 clean reads were retained. Among them, 128,885,514 reads 18 to 30 nt in length were selected for subsequent analysis.

### 2.2. Identification and Characterization of Salt-Responsive miRNAs in Tomatoes

In total, 244 miRNAs consisting of 135 known miRNAs and 109 putatively novel miRNAs were identified in this study (Appendix A). The lengths of the identified miRNAs were generally 21 or 24 nt (Figure 1A). The conserved miRNAs were classified into 65 families. Roughly 75% of miRNA families had more than one member. Among them, the three largest families, miR156, miR171, and miR482, contained seven members, and they were followed by the miR7981 family with six members (Figure 1B). There were 117 known miRNAs expressed in both M82 and *S. pennellii*. However ten (miR5304/6025/9469-3p/9472-5p/9476-5p/9477-5p/9478-5p/10529/10530/10537) and eight (miR169a/319d/394-3p/399b/5302b-3p/6023/10540/10542) known miRNAs were found to be expressed only in M82 or *S. pennellii*, respectively. For the novel miRNAs, 32 miRNAs were detected in both cultivars, while 37 and 40 miRNAs were found only in M82 or *S. pennellii*, respectively (Figure 1C). To identify the differentially expressed miRNAs (DE-miRNAs), we compared the expression levels of miRNAs between control and salt-treated samples. In M82, the 145 DE-miRNAs that were significantly regulated (82 known miRNAs and 63 novel miRNAs) consisted of 20 up-regulated and 125 down-regulated miRNAs. In *S. pennellii*, the 53 DE-miRNAs that were significantly regulated under salt stress (19 known miRNAs and 34 novel miRNAs) consisted of 16 up-regulated and 37 down-regulated miRNAs (Figure 1D,E). Interestingly, all known DE-miRNAs (*p*-value ≤ 0.05) were significantly down-regulated under salt stress in M82 or *S. pennellii*. There were 13 known miRNAs down-regulated in both of the two cultivars, and 70 and 6 known miRNAs (sly-miR164a-3p/171e/171f/319a/394-5p/477-3p) were down-regulated only in M82 or *S. pennellii*, respectively. The detailed information on the DE-miRNAs is shown in Appendix A.

### 2.3. Prediction and Functional Category of the Putative Target Genes of Salt-Responsive miRNAs

By using the psRNATarget server, the putative target genes for the DE-miRNAs were predicted. There were 5615 and 2768 genes predicted to be targeted by the 145 and 53 miRNAs in M82 and *S. pennellii*, respectively (Appendix A). Previous studies have shown that some members of the TF families, such as AP2/EREBP [87], GRAS [88,89], MYB [90,91], MYC [92], bHLH [93], HSF [94], WRKY [95,96], NAC [97], MADS-box [98], and TCP [99], are involved in the responses of plants to salt stress. In this study, many DE-miRNAs were predicted to target salt-related TFs. For the DE-miRNAs that were only differentially expressed in *S. pennellii*, three GRAS family TFs were targeted by sly-miR171e, *MYB65/97* and *TCP4* were targeted by sly-miR319a, *HD-ZIP* was targeted by sly-miR394-5p, and sly-miR477-3p could target *bHLH*, *NAC71*, and *MYB36/62*. For the DE-miRNAs that were differentially expressed in both of the cultivars, sly-miR169e-3p was predicted to target *MADS-*box, while both *bHLH* and *WRKY33* could be targeted by sly-miR390a-3p, and sly-miR482e-5p could target GRAS family TFs. Some target genes of DE-miRNAs were also found to be associated with the plant hormone signal transduction and MAPK signaling pathways, such *ABI2/5*, *HK2/3* (histidine kinase), *PP2C*, *MAPKKK8* (MAPK/ERK kinase kinase), *ARF2/18* (auxin response factor), *CTR1*, and so forth.

The same miRNA can regulate different genes, and the same gene can be targeted by different miRNAs from the same or different miRNA families. For instance, three genes (Solyc04g076850.3 (*IAA9*), Solyc09g083280.3 (*IAA3*, *SHY2*), and Solyc12g007230.2 (*IAA27*, *PAP2*)) were predicted to be targeted by sly-miR390a-3p and sly-miR390b-3p, while Solyc11g006180.2 (ethylene insensitive 4 (*EIN4*)) could be targeted by sly-miR390a-5p and sly-miR390b-5p. Solyc02g071220.3 (*ARR9*, *ATRR4*) could be targeted by sly-miR156a/b/c/d-5p/e-5p. Solyc08g005050.3 (*MYC2*) and Solyc09g009490.3 (*ABI5*) could be targeted by sly-miR166a/b/c-3p, while sly-miR166c-5p was predicted to target Solyc02g068490.3 (responsive-to-antagonist 1 (*RAN1*)). This suggests that there might be a functional redundancy among the different members of one miRNA family, while different functions of different members of the same miRNA family might be related to nucleotide divergence.

To further investigate the functions of salt-responsive DE-miRNAs, Gene Ontology (GO) enrichment and Kyoto Encyclopedia of Genes and Genomes (KEGG) pathway analyses were carried out for the target genes of DE-miRNAs. The GO analysis results showed that all target genes were classified into three categories (biological process (BP), molecular function (MF), and cellular component (CC)). The GO term “response to stimulus” (GO:0050896) was the most predominant term in the BP category, which contained 1767 and 884 genes in M82 and *S. pennellii*, respectively. The other GO terms are shown in Figure 2A,B. Based on the KEGG analysis, the target genes of the DE-miRNAs were, where significant, classified into the metabolism, genetic information processing, environmental information processing, cellular processes, and organismal systems pathways in M82, while the targeted genes in *S. pennellii* were classified into four of the pathways above, but not cellular processes. Some putative target genes of the two cultivars were involved in the mismatch repair, plant–pathogen interaction, plant hormone signal transduction, and MAPK signaling pathways (Figure 2C,D).

### 2.4. Validation of Salt-Responsive miRNAs

Quantitative real-time PCR (qRT-PCR) was utilized to confirm the quality of high-throughput sequencing and the expression patterns of salt-responsive miRNAs. Three known and three novel miRNAs were randomly selected. Similar tendencies were observed for the qRT-PCR and RNA-seq results for the expression of the selected miRNAs, suggesting that the results of the Illumina sequencing were reliable. The expression levels of the four miRNAs (miR397-3p, miR390a-3p, novel_mir_015, and novel_mir_081) were dramatically reduced in both M82 and *S. pennellii*. This suggests that the expression levels of salt-responsive miRNAs significantly varied under salt stress (Figure 3). The expression levels of miR319a and novel_mir_014 were slightly reduced in both M82 and *S. pennellii*. However, miR319a was down-regulated in *S. pennellii* and novel_mir_014 was up-regulated in M82.

### 2.5. Identification and Characterization of Salt-Responsive mRNAs and CircRNAs

There were twelve qualified libraries from the control (M82-CK and P-CK) and salt-treatment (M82-T and P-T) groups sequenced (three biological replicates per sample were assayed). After removing the primers and low-quality sequences, a total of 588.98 G high-quality sequences remained (Table 2). Through transcriptome analysis, 25,255 mRNAs were identified across twelve libraries, including 517 and 20 mRNAs that were expressed only in M82 or *S. pennellii*, respectively (Figure 4A). In addition, 4324 circRNAs were identified using the find_circ software. Among them, 2495 and 1509 circRNAs were identified only from M82 or *S. pennellii*, respectively. Only 320 circRNAs were detected in both of the cultivars (Figure 5).

The read counts of mRNAs and circRNAs were then normalized by TPM value (Appendix A). In total, there were 5174 DE-mRNAs and 107 DE-circRNAs found in the roots of the two cultivars under salt stress (Figure 4B,C and Figure 5 and Appendix A). Among them, sly_circ_3635 and sly_circ_3485 were up-regulated in M82 and *S. pennellii*, respectively, while sly_circ_3614 was down-regulated in both cultivars. In M82, 1281 mRNAs and 27 circRNAs were up-regulated and 2189 mRNAs and 20 circRNAs were down-regulated, while in *S. pennellii*, 1077 mRNAs and 33 circRNAs were up-regulated and 2274 mRNAs and 30 circRNAs were down-regulated. From the results presented above, it seems that more genes were down-regulated than up-regulated. In addition, 1647 DE-mRNAs were co-expressed in the two cultivars, including 58 mRNAs with opposite expression trends and 1589 mRNAs with similar expression trends. The genes that had similar expression patterns might have been related to the shared salt stress-responsive mechanisms of the two cultivars.

### 2.6. GO and KEGG Analysis of Differentially Expressed mRNAs

To investigate the differences in salt-responsive mRNAs between M82 and *S. pennellii*, three mRNAs sets were selected to perform the GO and KEGG enrichment analyses, respectively (set A: the mRNAs that were only differentially expressed in M82; set B: the mRNAs that were only differentially expressed in *S. pennellii*; set C: the mRNAs that were differentially expressed in both cultivars). The GO enrichment analysis showed that there were 88, 103, and 133 GO terms enriched in set A, set B, and set C, respectively. As shown in Figure 4, the most significant GO term in the BP category contained in set A and set B was “oxidation–reduction process” (GO:0055114), which contained 153 (59 up-regulated and 94 down-regulated) and 145 (49 up-regulated and 96 down-regulated) mRNAs in M82 and *S. pennellii*, respectively. In M82, “transport” (GO:0006810), “response to chemical” (GO:0042221), and “response to oxidative stress” (GO:0006979) were also enriched. ”Response to stimulus” (GO:0050896), “photosynthesis” (GO:0015979), and “carbohydrate metabolic process” (GO:0005975) were enriched in *S. pennellii*. For the MF category, “oxidoreductase activity” (GO:0016491), “tetrapyrrole binding” (GO:0046906), and “heme binding” (GO:0020037) were enriched in both cultivars. The KEGG results showed that several metabolic pathways were significantly enriched in the two cultivars. In M82, the nitrogen metabolism (*p*-value = 2.97 × 10^−8^), phenylpropanoid biosynthesis (*p*-value = 1.66 × 10^−6^), and biosynthesis of secondary metabolites (*p*-value = 2.17 × 10^−4^) categories were significantly enriched. The photosynthesis-antenna proteins (*p*-value = 3.28 × 10^−17^), biosynthesis of secondary metabolites (*p*-value = 5.09 × 10^−15^), and phenylpropanoid biosynthesis (*p*-value = 2.05 × 10^−8^) categories were significantly enriched in *S. pennellii*.

For the mRNAs of set C, the GO analysis results showed that “cell wall organization or biogenesis” (GO:0071554), “microtubule-based process” (GO:0007017), and “movement of cell or subcellular component” (GO:0006928) were significantly enriched (Appendix A). There were 45 mRNAs contained under the “cell wall organization or biogenesis” term and, among them, XTH5 was down-regulated and up-regulated in M82 and *S. pennellii*, respectively. The other 44 mRNAs exhibited similar expression patterns (12 up-regulated and 32 down-regulated) in the two cultivars. There were 34 mRNAs contained under the “microtubule-based process” term and they were significantly down-regulated in both cultivars. The KEGG enrichment results for set C showed that the DNA replication (*p*-value = 2.96 × 10^−6^), phenylpropanoid biosynthesis (*p*-value = 1.27 × 10^−5^), phenylalanine metabolism (*p*-value = 1.49 × 10^−5^), and plant hormone signal transduction pathways (*p*-value = 4.27 × 10^−5^) were significantly enriched (Appendix A).

### 2.7. Construction of Salt-Responsive circRNA–miRNA–mRNA Networks

The ceRNA hypothesis revealed a new mechanism of interaction between several RNAs. It is known that miRNAs can cause gene silencing through translational repression of mRNAs and that ceRNAs can regulate the expression of the corresponding gene by binding to miRNAs competitively. In this study, the ceRNA regulatory network, with the miRNAs as the core elements and mRNAs and circRNAs as target genes, was constructed by searching for circRNAs and mRNAs with the same miRNA binding sites. In order to investigate the function of the ceRNA regulatory network, several circRNA–miRNA–mRNA networks were constructed by integrating the expression profiles and regulatory relationships between mRNAs, circRNAs, and miRNAs.

The ceRNA regulatory network of M82 consisted of 281 miRNA–mRNA pairs and 33 miRNA–circRNA pairs with negative correlations between RNA pairs; among them, only sly_circ_871 was down-regulated (Figure 6A,B) (Appendix A). The ceRNA regulatory network of *S. pennellii* consisted of 284 miRNA–mRNA pairs and 60 miRNA-circRNA pairs (Appendix A).

GO and KEGG analyses of the targeted DE-mRNAs were then performed to reveal the potential functions of the miRNAs and circRNAs contained in the ceRNA networks. Compared to the GO enrichment results for the M82 line, 26, 7, 10, and 8 DE-mRNAs in *S. pennellii* were enriched for “external encapsulating structure organization” (GO:0045229), “suberin biosynthetic process” (GO:0010345), “defense response to insect” (GO:0002213), and “cell wall pectin metabolic process” (GO:0052546) especially. However, 40, 46, 34, and 30 DE-mRNAs in the M82 line were enriched for “response to lipid” (GO:0033993), “response to acid chemical” (GO:0001101), “cellular response to hormone stimulus” (GO:0032870), and “hormone-mediated signaling pathway” (GO:0009755), respectively. Furthermore, two GO terms, “response to oxygen-containing compound” (GO:1901700) and “response to endogenous stimulus” (GO:0009719), were significantly enriched in both cultivars (Appendix A).

Based on the KEGG results, two pathways (plant hormone signal transduction and biosynthesis of secondary metabolites) were enriched in both cultivars (Appendix A). Furthermore, there were three, six, and two mRNAs involved in the taurine and hypotaurine metabolism (*p*-value = 1.64 × 10^−4^), MAPK signaling (*p*-value = 2.04 × 10^−3^), and nitrogen metabolism pathways (*p*-value = 2.5 × 10^−2^) in M82, respectively. Moreover, 38 mRNAs were involved in the metabolic pathways (*p*-value = 5.3 × 10^−4^), and some other stress-related pathways were also enriched in *S. pennellii*. For instance, five mRNAs were predicted to be enriched in both the fatty acid degradation (*p*-value = 2.49 × 10^−4^) and photosynthesis pathways (*p*-value = 1.06 × 10^−3^). Six and four mRNAs were enriched in the amino sugar and nucleotide sugar metabolism (*p*-value = 1.40 × 10^−3^) and arginine and proline metabolism (*p*-value = 3.18 × 10^−3^) pathways, respectively. Solyc03g093650.3 and Solyc11g065720.2 were involved in the ABC transporters pathway (*p*-value = 3.1 × 10^−2^); Solyc03g111120.3, Solyc06g035960.3, and Solyc03g025720.3 were involved in the glyoxylate and dicarboxylate metabolism pathway (*p*-value = 3.2 × 10^−2^); Solyc03g097500.3 and Solyc06g076800.3 were involved in the cutin, suberine, and wax biosynthesis pathway (*p*-value = 3.66 × 10^−2^); Solyc09g091470.3, Solyc01g095745.1, and Solyc01g095740.3 were involved in the fatty acid metabolism pathway (*p*-value = 3.85 × 10^−2^); Solyc04g050930.3 and Solyc01g108220.3 were involved in the carotenoid biosynthesis pathway (*p*-value = 4.67 × 10^−2^); and five mRNAs (Solyc07g008360.2, Solyc10g084240.2, Solyc02g084790.3, Solyc02g030480.3, and Solyc02g087190.1) were involved in the phenylpropanoid biosynthesis pathway (*p*-value = 4.76 × 10^−2^).

Based on the above results, the GO term “suberin biosynthetic process” and the KEGG plant hormone signal transduction pathway were selected to further examine the function of salt stress-responsive ceRNA networks. By analyzing the ceRNA networks of plant hormone signal transduction, it was found that all the target mRNAs in M82 were up-regulated. There were two DE-mRNAs, Solyc07g008250.3 (*EBF1*) and Solyc07g040990.3 (*ABI2*), targeted by novel_mir_001, and ABI2 could be targeted by novel_mir_027 in *S. pennellii* especially. Novel_mir_001 was predicted to target sly_circ_2467 and sly_circ_772 in M82 and *S. pennellii*, respectively. The expression patterns of novel_mir_025 showed opposite trends in the two cultivars (down-regulated in M82 and up-regulated in *S. pennellii*). Novel_mir_025 was predicted to target sly_circ_1857 and Solyc07g006860.3 (xyloglucan endotransglycosylase 6 (*XTR6*)) in M82, while they were predicted to target sly_circ_1475, sly_circ_3512, sly_circ_3614, and Solyc03g093130.3 (*XTR6*) in *S. pennellii*. Beyond that, 11 DE-mRNAs were enriched only in M82. Among them, novel_mir_007 was predicted to target Solyc11g006180.2 (*EIN4*), sly_circ_154, and sly_circ_309. Novel_mir_017 was predicted to target Solyc12g099230.2 (pectin lyase-like superfamily protein) and sly_circ_3635. sly-miR1917 was predicted to target Solyc06g053710.3 (EIN4 ethylene sensor), Solyc12g096980.2 (*IAA13*), sly_circ_2508, and sly_circ_3635. Novel_mir_037 was predicted to target Solyc04g078840.3 (*ABF2*, *AREB1*), Solyc07g063850.3 (*DFL1*, *GH3.6*, auxin-responsive GH3), and sly_circ_3552. There were also five DE-mRNAs enriched only in *S. pennellii* (Figure 7A,B). The RNA pairs contained sly_circ_4277-novel_mir_051-Solyc08g005050.3 (*MYC2*), sly_circ_3020-novel_mir_096-Solyc10g079460.1 (BOP2 ankyrin repeat family protein BTB/POZ domain-containing protein), and sly_circ_2710-novel_mir_109-Solyc11g011030.2 (*JAZ2*). sly-miR9474-3p was predicted to target Solyc05g052980.3 (*PP2CA*), sly_circ_1987, and sly_circ_4277.

By analyzing the ceRNA networks of the GO term “suberin biosynthetic process”, it was found that seven genes that contained Solyc08g008610.3 (alpha/beta-hydrolases), Solyc10g009240.3 (*KCS1*, 3-ketoacyl-CoA synthase 1), Solyc03g097500.3 (HXXXD-type acyl-transferase), Solyc09g005970.1 (ABC-2 type transporter), Solyc10g078280.3 (*CYP94D2*), Solyc06g076800.3 (*CYP86A*), and Solyc03g111300.1 (*CYP94B3*) were enriched. Novel_mir_020 was predicted to target Solyc10g009240.3 (*KCS1*), Solyc03g097500.3 (HXXXD-type acyl-transferase), and three circRNAs (sly_circ_1470, sly_circ_2651, and sly_circ_3601). In this network, novel_mir_027 and novel_mir_051 were also predicted to target Solyc06g076800.3 (*CYP86A1*, cytochrome P450, family 86, subfamily A, polypeptide 1) and two mRNAs (Solyc03g111300.1 (*CYP94B3*) and Solyc04g078280.3), respectively. sly-miR477-3p and novel_mir_078 were predicted to target Solyc09g005970.1 (ABC-2 type transporter) and Solyc08g008610.3 (alpha/beta-hydrolases), respectively. As described above, this complicated network showed that these circRNAs might play regulatory roles in the suberin biosynthetic process through the seven miRNAs and their target genes under salt stress in *S. pennellii*.

## 3. Discussion

In recent years, great progress had been made in the research on the physiological mechanisms of salt tolerance in plants [100,101,102,103,104]. With the rapid development of molecular biology and next-generation high-throughput sequencing technologies, some salt-related genes have already been used in transgenic research [105,106,107,108,109,110]. Therefore, it is important to develop methods to analyze the differences in salt-responsive mechanisms between wild plants (salt-tolerant) and cultivated plants (salt-sensitive) from the perspective of genomics, and then excavate some new salt-tolerant genes and establish complete networks of salt-tolerance mechanisms. In this study, we identified the salt-responsive DE-miRNAs, DE-mRNAs, and DE-circRNAs in the roots of cultivated and wild tomatoes. To analyze the differences in the salt-responsive mechanisms between the two cultivars, we selected the RNA pairs (circRNA–miRNA and miRNA–mRNA pairs) and established several ceRNA networks according to ceRNA theory.

### 3.1. Analysis of Salt-Responsive mRNAs in Tomatoes

By comparing the DE genes of the two cultivars, we found that some genes related to salt stress showed opposite expression patterns in the two cultivars, including Solyc04g005810.3 (thioredoxin 2 (*TRX2*)), Solyc05g015300.3 (responsive to dessication 22 (*RD22*)), Solyc11g016930.1 (leucine-rich repeat protein kinase (LRR)), Solyc11g013890.1 (metallothionein 2A (*MT2A*)), and other genes. The expression levels of these genes were significantly up-regulated in *S. pennellii* and down-regulated in M82. TRX has been reported to play a crucial role in mediating the redox state in living cells by transferring the putative Trxf from non-secretor mangrove Kandelia candel to tobacco, and the KcTrxf transgenic lines showed higher salt tolerance than wild-type tobacco [111]. Agrobacterium transformation experiments for the *RD22* gene in tobacco model plants (*Nicotiana benthamiana*) found that the salt tolerance of the transgenic plants was enhanced, which involved osmotic regulation strategies and protective effects [112]. LRRs played important roles in signal perception and activation of defense responses. Studies in tobacco [113] and *Medicago truncatula* (*M. truncatula*) [114] showed that the expression level of LRR was up-regulated under salt stress. In Arabidopsis, the overexpression of the *MT2A* gene could enhance the stress resistance by maintaining chlorophyll, increasing the K^+^/Na^+^ ratio and proline content and decreasing ROS level under salt stress [115]. These genes were only up-regulated in *S. pennellii*, which might be one of the reasons for the high salt tolerance of wild tomatoes.

In addition, 1312 salt-responsive genes were enriched in M82 and 241 genes were directly involved in the response to salt stress. In addition to the 618 salt-responsive genes that were shared with M82, 634 genes were also enriched in *S. pennellii*. Furthermore, we also found that the phenylpropane biosynthesis, plant hormone signal transduction, MAPK signal transduction and biosynthesis of secondary metabolite pathways were the most abundant metabolic pathways across the two cultivars. In M82, several pathways, such as the nitrogen metabolism, plant–pathogen interaction, and fatty acid biosynthesis pathways, as well as others, were significantly enriched. In *S. pennellii*, photosynthesis, flavonoid biosynthesis, carbon metabolism, carotenoid biosynthesis, and starch and sucrose metabolism were significantly enriched. We speculated that these differential metabolic pathways were related to the different signal transduction patterns produced by the two cultivars under salt stress. This result showed that there were great differences in the mechanisms of the two cultivars under salt stress.

However, the mechanisms of salt tolerance in tomatoes were only examined at the mRNA level and not at the entire transcriptome level in the above studies. More evidence has shown that ncRNAs, such as miRNAs [43,45,56] and circRNAs [82], can also play a significant role in the process of plant salt-tolerance. The function of miRNAs capable of regulating the gene expression level at the post-transcriptional level under salt stress in tomatoes has been studied in detail before [116]. However, there are few reports related to salt-responsive ceRNA networks in tomatoes.

In this study, we identified salt-responsive DE-miRNAs, DE-mRNAs, and DE-circRNAs in the roots of cultivated and wild tomatoes under salt treatment and established several putative salt-responsive ceRNA networks based on the potential interaction relationships between RNAs. This will deepen our understanding of the mechanisms of salt tolerance.

### 3.2. miRNAs Play Crucial Roles in Tomatoes under Salt Stress

Over the long process of evolution, plants have evolved a complex regulatory network to deal with environmental stresses. miRNAs play a critical role in this network at transcriptional and post-transcriptional levels. In plants, miRNAs have been found to be involved in plant responses to abiotic stress [64,117,118], growth and development [119,120], and signal transduction [121,122]. Like miR168, miR171, and miR396 were found to respond to salt, drought, and cold stress, miR165, miR167, miR319, and miR393 have been found to be induced by salt stress [60].

In this study, we profiled the miRNAs in M82 (salt-sensitive tomato) and *S. pennellii* (salt-tolerant tomato) roots expressed under salt stress and some salt-responsive miRNAs were identified. In total, 135 conserved miRNAs and 109 putative novel miRNAs were detected. The conserved miRNAs could be classified into 65 miRNA families. Among them, 117 conserved miRNAs were expressed in both two cultivars, while only 10 and 8 known miRNAs were specific for M82 and *S. pennellii*, respectively. There were 88 conserved miRNAs that were significantly down-regulated under salt treatment across two cultivars, with 69 and 6 miRNAs inhibited in M82 and *S. pennellii*, respectively. Only 13 miRNAs were shared by two cultivars. By comparing the results from this study and those from previous studies, we found that most of these salt-responsive miRNAs or their homologs in other plant species have been identified as salt-stress-regulated miRNAs, which suggests that these miRNAs may be part of an evolutionarily conserved stress-response mechanism across different species.

The down-regulation of miR166 and up-regulation of target gene *HD-ZIP* in salt-tolerant soybean might enhance salt tolerance [123,124]. In rice and Arabidopsis, high salt stress could significantly induce the members of the miR169g family (miR169g and miR169n-o) to participate in the salt response process by targeting the NF-YA TF. However, sly-miR169e-3p was significantly down-regulated in both two cultivars, which might indicate that the members of the miR169 family had different responses to salt stress in different species [44]. In *Helianthus tuberosus* L, the expression level of miR390 was up-regulated under 100 mM NaCl treatment and down-regulated under 300 mM NaCl treatment, while miR390 could also be induced in poplar (*Populus* spp.) plants under salt stress [125,126]. The two members of miR390 families (sly-miR390a-3p and sly-miR390a-5p) were down-regulated in both M82 and *S. pennellii* under 200 mM NaCl treatment in this study, indicating that the conservation of salt-responsive miRNAs among different species might have been related to the salt concentration and that there was a certain cultivar specificity among the different cultivars. It has been reported that the miR396 and miR397 families are salt-inhibited miRNAs in other species, which is consistent with the results in this study [127,128,129]. miR408 was induced in Arabidopsis [130] and cotton [131], while it was significantly inhibited in rice [132], *M. truncatula* [133], and radish [134] under salt treatment. This showed that the salt-responsive mechanism with miR408 in tomato might be similar to that in rice, *M. truncatula,* and radish. miR482 was also found to be inhibited in sweet potato and bread wheat [135,136]. In soybean, the expression levels of miR482b/d/e were significantly decreased, which is consistent with that the levels of miR482b and miR482e-5p in this study [137]. There was even a hitherto unrecognized association between two miRNAs (sly-miR9474-3p and sly-miR9476-3p) and salt stress, but they could be inhibited in tomatoes under drought stress [138]. There might be a common regulatory process or signal crossover between high salt stress and drought stress. These miRNAs also negatively regulate salt- and drought-stress signaling. In tomato, sly-miR9479-3p has been reported to target cytochrome P450, MYB, and bHLH under low temperatures, but there are no studies of sly-miR9479-3p involvement in salt stress tolerance to date. Our study showed that there might be huge differences in miRNA responses to salt stress between different species and the differences might be related to the degree of salt stress. The DE-miRNAs might explain the different salt sensitivities of the two tomato cultivars. The similarly regulated miRNAs might represent the basic mechanism of adaptation to salt stress during the evolution of tomatoes. The different members of the same miRNA family might play different roles in the response to salt stress.

In addition, we found six miRNAs (sly-miR164a-3p, sly-miR171e, sly-miR171f, sly-miR319a, sly-miR394-5p, and sly-miR477-3p) that were specific to *S. pennellii* and were probably related to salt tolerance. Solyc06g060620.3 (*NRT1.1*) was predicted to be targeted by sly-miR164a-3p and up-regulated in *S. pennellii*, while previous research found that the expression level of *NRT1.1* was down-regulated with or without NaCl treatment (100 mM) at seven days. This indicated that *S. pennellii* might accumulate more NO_3_^−^ early in the response to salt stress, and NO_3_^−^ has been proved to enhance the salt tolerance of plants as osmotic regulators [139,140,141,142]. The target TFs of miR171 were also identified as being involved in the further regulation of gene expression and signal transduction that probably functioned in stress responses [143,144,145]. The miR319 family had been confirmed to be involved in salt, heat, and cold stresses in tomatoes by targeting TCP TFs [146,147,148], and miR319 could also mediate ethylene biosynthesis and signal transformation by inhibiting the expression of its target gene *PvPCF5* to improve the salt tolerance of C4 plant switchgrass [149]. miR394 and its target gene *LCR* were confirmed to be involved in Arabidopsis salt- and drought-stress responses in an abscisic acid-dependent manner [150,151]. The pentatricopeptide repeat (PPR) protein family was reported to provide a signaling link between mitochondrial electron transport and regulation of stress and hormonal responses in Arabidopsis [117,152] and was targeted by the miR477 family, which was involved in rice under salt stress [153]. These results confirm the important roles of the six miRNAs in *S. pennellii* under salt stress and shed new light on the causes of the high salt tolerance of wild tomatoes.

### 3.3. ceRNA Networks Could Shed New Light on the Regulatory Roles of ncRNAs

The mechanisms of salt-response processes in plants are very complex, involving a variety of genes and reaction components. Although researchers have studied the roles of circRNAs, miRNAs, and other ncRNAs in the maintenance of plant response to stress extensively [71,76,154], the salt-responsive ceRNA (circRNA–miRNA–miRNA) regulatory network has not been widely constructed in tomatoes to date. To explore the function of ceRNA networks in tomatoes in the response to salt treatment and the specific salt-tolerance mechanism of the wild tomato *S. pennellii*, we constructed several salt-responsive ceRNA (circRNA–miRNA–mRNA) networks in M82 and *S. pennellii*.

By comparing the networks of two cultivars, we found four shared miRNAs (sly-miR397-5p, novel_mir_001, novel_mir_025, and novel_mir_081). Among them, novel_mir_025 was down-regulated in M82 and up-regulated in *S. pennellii*, while the other three miRNAs were down-regulated across the two cultivars. Previous studies have shown that the laccase enzymes (LAC) belong to the blue copper oxygenase family and they were proved to be the target genes of miR397 in Arabidopsis and rice [155,156]. The over-expression of *LAC* could also improve biotic stress tolerance in plants [157]. miR397 was found to target several members of the LAC family (*LAC2*, *LAC4*, and *LAC17*), which were copper-containing enzymes that could catalyze the oxidation of phenolic compounds and the concomitant reduction of oxygen to water [158]. Laccases were found to be involved in a diverse range of functions related to defense and cell-wall lignification. In this study, Solyc06g082260.3 (*LAC3*), Solyc06g050530.3 (*LAC12*), and Solyc06g076760.2 (laccase/diphenol oxidase) were putative target genes of sly-miR397-5p and sly_circ_1172 and sly_circ_2651 were predicted to bind to sly-miR397-5p in M82 and *S. pennellii*, respectively. In addition, relatively higher expression levels of these three genes and sly-miR397-5p were detected in *S. pennellii* compared to M82. This indicated that the expression of the *LAC* gene was involved in salt tolerance in tomato and that sly_circ_2651 might increase the expression levels of *LAC* by binding to sly-miR397-5p, thereby contributing to the salt tolerance of *S. pennellii* wild tomato.

Based on the GO enrichment analysis results of the target genes, two GO terms (“response to oxygen-containing compound” (GO:1901700) and “response to endogenous stimulus” (GO:0009719)) were shared by two ceRNA networks. Among the genes contained under these terms, Solyc02g089160.3 (*BR6OX1*) and Solyc11g056650.2 (ABA-responsive kinase substrate 1 (*FBH3*/*AKS1*)) showed similar expression levels and patterns across the two cultivars. Solyc06g076760.2 (*LAC4*), Solyc06g050530.3 (*LAC12*), and Solyc02g092860.3 (*CYP81D8*) were expressed at higher levels in *S. pennellii* compared to in M82. Brassinolide has been confirmed to be ubiquitous in the plants. Under salt-stress conditions, it can affect the growth and development of plants in a variety of ways and improve the ability of plants to resist stresses [20]. As one of the synthetic genes of brassinolide, Solyc02g089160.3 (*BR6OX1*, *CYP85A1*) was up-regulated in both two cultivars and was predicted to be targeted by novel_mir_017 and novel_mir_027 in M82 and *S. pennellii*, respectively. sly_circ_3635 and seven circRNAs (sly_circ_727/1470/2702/2703/2651/3601/3778) were predicted to be targeted by novel_mir_017 and novel_mir_027, respectively. This indicated that tomatoes might regulate the biosynthesis of BR by regulating the expression of the *BR6OX1* gene under salt stress, thereby affecting the plant’s resistance to stress. FBH3/AKS1 is a transcriptional activator bound to the promoter of KAT1 (inwardly rectifying K^+^ channels) that is involved in the stomatal opening. *CYP81D8* is a general oxidative stress-response marker gene. sly_circ_936 and sly_circ_1470/2651/3601 were predicted to be targeted by sly-miR156e-5p and novel_mir_020 in M82 and *S. pennellii*, respectively, while *CYP81D8* was predicted to be targeted by sly-miR156e-5p and novel_mir_020. This showed that sly_circ_1470/2651/3601 might regulate the expression level of *CYP81D8* by binding to novel_mir_020, which would in turn have contributed to the salt tolerance of the *S. pennellii* wild tomato.

We also found seven genes that were enriched in the suberin biosynthetic process (GO:0010345) in *S. pennellii* and we constructed a suberin-associated ceRNA network. Except for Solyc03g111300.1 (*CYP94B3*) and Solyc10g078280.3 (*CYP94D2*), the other five genes were expressed at higher levels in *S. pennellii* compared to in M82. Suberin is a kind of lipophilic biopolyester and suberized cell walls act as barriers to limit the transport of water and nutrients and prevent invasion [159]. A previous study proved that the Arabidopsis cytochrome P450 *CYP86A1* and *CYP86B1* could encode a fatty acid ω-hydroxylase involved in suberin monomer biosynthesis, while only co-overexpression of *GPAT5* and either *CYP86A1* or *CYP86B1* could format long- and very-long-chain ω-hydroxyacids and DCAs in the cutin layer [160,161,162]. Under salt stress, the aliphatic suberin could reduce the backflow of water to the medium and the Arabidopsis mutant *cyp86a1* could accumulate fewer K^+^ in shoots than wild-type [163]. In this study, although Solyc04g011600.3 (glycerol-3-phosphate acyltransferase 5 (*GAPT5*)) and Solyc06g076800.3 (*CYP86A1*) were both up-regulated across the two cultivars, the expression levels of the two genes were over tenfold higher in *S. pennellii* compared to M82. Solyc06g076800.3 and seven circRNAs (sly_circ_727/1470/2702/2703/2651/3601/3778) were predicted to be targeted by novel_mir_027. This result showed that the seven circRNAs might regulate the expression level of *CYP86A1* by binding to novel_mir_027 and contribute to the biosynthesis of suberin in *S. pennellii* under salt stress. According to previous studies, *CYP94B1/2/3* and *CYP94C1* are responsible for catalyzing the sequential ω-oxidation of JA-Ile in a semi-redundant manner. Among them, CYP94B enzymes and *CYP94C1* might play primary and minor roles in the JA response process, respectively [164,165,166,167]. However, in this study, Solyc03g111300.1 (*CYP94B3*) was down-regulated across the two cultivars, which might have been related to the concentration of sodium chloride solution [168]. sly_circ_4277 and Solyc03g111300.1 were predicted to be targeted by novel_mir_051, and this showed that sly_circ_4277 might regulate the JA response process by binding to novel_mir_051.

Based on the KEGG enrichment analysis results, two pathways (plant hormone signal transduction and biosynthesis of secondary metabolites) were shared by two cultivars. Beyond this, the taurine and hypotaurine metabolism, MAPK signaling, and nitrogen metabolism pathways were enriched in M82, while the fatty acid degradation, photosynthesis, amino sugar and nucleotide sugar metabolism, ABC transporters, and phenylpropanoid biosynthesis pathways were significantly enriched in *S. pennellii*. The circRNA sly_circ_1857 might be a salt-responsive circRNA and act as a miRNA sponge by binding to sly-miR396a-5p and sly-miR396b, which in turn could regulate the expression of Solyc04g025530.3 ( glutamate decarboxylase 4 (*GAD4*)). GAD4 has already been verified to be a rate-limiting enzyme involved in GABA biosynthesis and validated as a real target gene of miR396 by degradome sequencing in a previous study [169]. The expression level of Solyc06g053710.3 (*EIN4*) was predicted to be regulated by sly_circ_2508/3635 through binding to sly-miR1917, while sly-miR1917 has been verified to be involved in the miRNA-mediated ethylene network in solanaceous species [170]. However, no study has shown that *EIN4* is targeted by miR1917 so far [171].

A previous study has shown that *ABI2* can interact with *SOS2* and cause decreased tolerance to salt stress and ABA sensitivity in plants [172]. In this study, the up-regulated circRNA sly_circ_2467 might have upregulated the expression level of Solyc07g040990.3 (*ABI2*) by binding to novel_mir_001 and played a negative role in salt-stress adaptation. As a major rate-limiting enzyme involved in ethylene biosynthesis, the expression level of *ACS2* has been reported to be induced by *MPK3* and *MPK6* via *WRKY33* [173]. However, the expression level of *MPK3* was down-regulated across the two cultivars, while the expression levels of *MPK6* and *WRKY33* exhibited no significant changes in this study. This showed that sly_circ_2508/3635 might induce the expression level of Solyc01g095080.3 (*ACS2*) by binding to sly-miR1917 under salt stress in M82 [174]. A higher capacity to assimilate nitrogen has been found in transgenic tobacco (*Nicotiana tabacum* L.) that overexpressed *NADH-GOGAT* (NADH-dependent glutamate synthase) [175]. miR397 has been found to be involved in N and Cu homeostasis and down-regulated under nitrogen starvation conditions in maize, and miR166 has also been found to be down-regulated [176,177]. In this study, Solyc03g083440.3 (*NADH-GOGAT*) was predicted to be targeted by sly-miR166c-5p, sly-miR397-5p, and sly-miR9472-5p, while sly_circ_1172 was predicted to bind to the above three miRNAs. This showed that sly_circ_1172 and the three miRNAs might be involved in the ammonium assimilatory pathway in M82 under salt stress [178].

The circRNAs sly_circ_2651/3778 and Solyc09g091470.3 (peroxisomal 3-ketoacyl-CoA thiolase 3 (*KAT2/PED1/PKT3*)) were predicted to be targeted by the miRNA novel_mir_055, while *KAT2* has already been verified to regulate the ABA signaling pathway positively, and *WRKY40* could repress the expression of *KAT2*. ABA could inhibit the expression of *WRKY40* by promoting the ABAR–WRKY40 interaction; however, there was no significant change in the expression levels of *KAT2* between the *cch* mutant (a mutant allele of the *ABAR* gene) and wild-type plants [179]. This showed that sly_circ_2651/3778 and novel_mir_055 might regulate *KAT2* expression, except for *WRKY40*, and play a positive regulatory role in ABA signal transduction under salt stress in *S. pennellii* [180].

Our research showed that specific circRNAs might act as sponges to bind to miRNAs, which in turn regulate the transcription of target genes in tomatoes under salt stress. In this study, the roots of cultivated tomato (M82) and wild tomato (*S. pennellii*) were sampled for whole transcriptome analysis. The salt-responsive DE-miRNAs, DE-circRNAs, and DE-mRNAs in tomatoes were integrated and salt-responsive ceRNA regulatory networks were constructed for the first time. By analyzing the ceRNA networks of the two cultivars, regulatory processes that might be related to the high salt tolerance of wild tomatoes were also proposed. These results provide support for further understanding of the potential functions and mechanisms of tomato salt-responsive miRNAs and circRNAs under salt stress.

## 4. Materials and Methods

### 4.1. Plant Materials and Preparation

Seeds of the cultivated tomato M82 (moderately salt-sensitive) and the wild tomato *S. pennellii* (salt-tolerant) were planted in pots with a mixture of vermiculite and perlite and placed in a growth chamber under a long-day environment (16 h day/8 h night), with a light intensity of 100 µmol m^2^/s, a temperature of 25 °C, and a relative humidity of 20–30%. Plants were cultivated for six weeks (Hoagland solution was provided on time every two weeks). The control group and treatment group were grown in the same environment, and the plants in the treatment group were treated with 200 mM NaCl. All root samples were collected from control and treated plants at 0 h and 12 h after external salt treatment, frozen in liquid nitrogen, and stored at −80 °C for subsequent RNA extraction and high-throughput sequencing.

### 4.2. Small RNA Sequencing and Identification of DE-miRNAs

Following the manufacturer’s protocol, an RNA Extraction Kit (Takara, Dalian, China) was used to extract total RNA. The purity, concentration, integrity, and genomic DNA contamination of the RNA samples were detected using a NanoDrop 2000 spectrophotometer (NanoDrop, Wilmington, DE, USA) to ensure the eligibility of the samples. An amount totaling 1.5 ug was taken as the starting amount for the RNA sample, and the volume was replenished to 6 μL with water and the libraries were constructed using a small RNA Sample Pre Kit (Illumina, San Diego, CA, USA). After the libraries were constructed, a Qubit 2.0 Fluorometer (Thermo Fisher Scientific, Waltham, MA, USA) was utilized to detect the concentrations of the libraries, then the concentrations of the libraries were diluted to 1 ng/μL. An Agilent 2100 bioanalyzer (Agilent Technologies, California, CA, USA) was used to detect the insert size, and the qRT-PCR method was used to quantify the effective concentration of the library and ensure library quality accurately. Then, deep sequencing of libraries was performed using a HiSeq X Ten platform (Illumina, San Diego, CA, USA). The length of small RNA sequencing reads was 50 nt (single-end (SE)).

To ensure the accuracy of the downstream analysis, we used the following criteria for quality control of the raw data that were generated by sRNA sequencing: raw reads were filtered by eliminating adapter reads, low-quality reads, and tags without 3′ primer. The reads that were shorter than 18 nt and longer than 30 nt were also removed. The clean reads were compared with the sequences of precursor and mature miRNAs that were downloaded from miRBase (v22, http://www.mirbase.org/; accessed on 10 May 2021). Known and novel miRNAs were identified by miRDeep2 [181]. RNA-fold was utilized to analyze the structure of miRNAs. The R statistical package edgeR [182] was utilized to analyze the DE-miRNAs (|Log_2_(Foldchange)| ≥ 1, *p*-value ≤ 0.05).

### 4.3. Prediction and Annotation of Potential Target Genes of DE-miRNAs

The online analysis tool psRNAtarget (https://www.zhaolab.org/psRNATarget/; accessed on 20 May 2021) was utilized to predict the potential target genes of all DE-miRNAs that were regulated at the post-transcriptional or translational levels. The target gene library comprised all the transcripts of the tomato reference genome. The miRNA library comprised all the DE-miRNAs. V2 was selected as the version of the scoring mechanism, and the specific settings were as follows: the number of the best target gene candidates was set to 200; the expectation value was set to 4; the penalty score of the G-U pair was set to 0.5; the penalty score for other mismatches was set to 1; the extra weight in the seed region was set to 1.5; the seed region sequence length was set between 2 nt and 13 nt; the maximum number of mismatches allowed in the seed region was set to 2; the complementary regional assessment score was set to 19; and the penalty score for the opening gap was set to 2. The GO enrichment and KEGG pathway analyses of target genes were performed using agrigo (http://systemsbiology.cau.edu.cn/agriGOv2/) and KOBAS (http://kobas.cbi.pku.edu.cn/kobas3/), respectively (accessed on 22 May 2021). The threshold of significantly enriched GO terms and KEGG pathways were set as a *p*-value ≤ 0.05.

### 4.4. Validation of miRNAs by qRT-PCR

Expression profiles of six miRNAs (three known and three novel miRNAs) were randomly selected for validation by qRT-PCR. Three logical replicates were performed for each of the selected miRNAs. Primers were designed using Primer5 (Premier Biosoft International, Palo Alto, CA, USA) [183]. A *p*-value ≤ 0.05 was considered to be significant. The 2^−ΔΔCT^ method was utilized to calculate relative expression levels [184]. The primer sequences are listed in Appendix A.

### 4.5. Strand-Specific RNA Sequencing and Identification of Differentially Expressed mRNAs and circRNAs

For strand-specific libraries, whole-genome sequencing of tomato roots had been performed in our another preprint study [185]. A NanoDrop 2000 spectrophotometer (NanoDrop, Wilmington, DE, USA), Qubit 2.0 (Thermo Fisher Scientific, Waltham, MA, USA), Aglient 2100 (Agilent Technologies, California, CA, USA), and electrophoresis methods were utilized to detect the purity, concentration, integrity, and genomic DNA contamination of RNA samples and ensure the quality of all samples. Then, an Epicentre Ribo-zero kit (Epicentre, Madison, WI, USA) was utilized to remove rRNA, and a Fragmentation Buffer was added to randomly interrupt rRNA-depleted RNA. rRNA-depleted RNA was used as a template to synthesize the first cDNA strand with six-base random hexamers, and then buffer, dATP, dUTP, dCTP, dGTP, RNase H, and DNA polymerase I were added to synthesize the second cDNA strand. An AMPure XP beads kit (Beckman Coulter, Brea, CA, USA) was utilized to obtain the purified cDNA. End repair and addition of A were performed for purified double-stranded cDNAs, then the AMPure XP beads kit was utilized for fragment size selection (Beckman Coulter, Brea, CA, USA). Finally, the U chains were degraded and the cDNA libraries were obtained through PCR enrichment. After assessing the quality of the libraries, the Illumina sequencing platform was utilized to perform the deep sequencing (Illumina, San Diego, CA, USA).

The quality control of the original sequencing data was performed using Fastp [186] with low-quality reads and adapter sequences were deleted. Then, HISAT2 [187] was utilized to align the clean reads to the reference genome of tomato (sly3.0) [188]. For mRNA analysis, the transcripts of each sample were assembled and merged using Cufflink and Cuffcompare [189]. The known mRNAs were identified according to the genomic sequence annotation of tomato. For circRNA analysis, samtools was utilized to extract the unmapped sequences after the alignments, then the python script unmapped2anchors.py contained in the find_circ [86] software package was used to extract the anchor sequences from both sides of the extracted unmapped sequences. The output fastq files from the python script (unmapped2anchors.py) were aligned to the reference genome. Finally, the identification of circRNAs was performed by using find_circ software. The R statistical package DESeq2 [190] was utilized to calculate the TPM values for transcripts and circRNAs in each library and to identify the DE-mRNAs and DE-circRNAs across the two cultivars. The cut-off criteria for the DE-mRNAs and DE-circRNAs were set as |Log_2_(Foldchange)| ≥ 1 and a *p*-value ≤ 0.05. The GO and KEGG analyses of DE-mRNAs were performed using agrigo and KOBAS, respectively (accessed on 23 May 2021). The threshold was set as a *p*-value ≤ 0.05.

### 4.6. Construction of the circRNA–miRNA–mRNA Networks

The interactions of miRNA–mRNA and miRNA–circRNA pairs were predicted by using psRNAtarget (accessed on 23 May 2021). Then, the RNA pairs were filtered with the following principles. Firstly, RNA pairs with the same miRNA binding site were selected. Then, the false-positive RNA pairs were removed by screening for reverse expression trends between miRNAs and target RNAs. The expression trend of circRNA was the same as that of mRNA, but opposite that of miRNA. The ceRNA networks were constructed with the potential targets (DE-mRNAs and DE-circRNAs) and corresponding DE-miRNAs. The software Cytoscape (v3.7.1) [191] and R package ggalluvial [192] were utilized to visualize the ceRNA networks.

### 4.7. GO and KEGG Pathway Analyses of mRNAs in the CeRNA Networks

GO enrichment and KEGG pathway analyses of DE-mRNAs in the circRNA–miRNA–mRNA networks were performed by utilizing the website tools agrigo and KOBAS, respectively (accessed on 24 May 2021). The threshold of significant GO terms and KEGG pathways was set as a *p*-value ≤ 0.05.

## Figures and Tables

**Figure 1 ijms-22-12238-f001:**
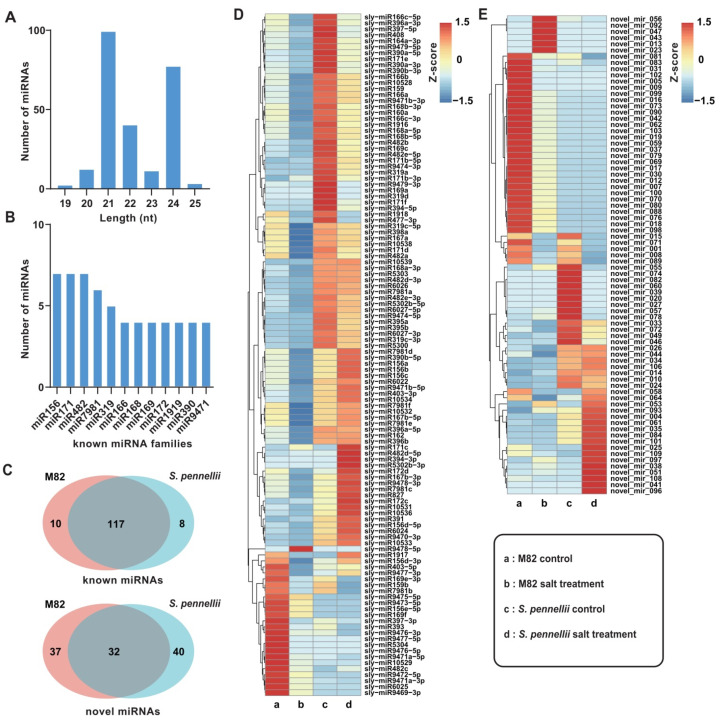
Identification and characterization of salt-responsive miRNAs. (**A**) The length distribution of salt-responsive miRNA read sequences. (**B**) The twelve conserved miRNA families had more than four members. (**C**) Venn diagram showing the intersection of miRNA expression in M82 and *Solanum pennellii* (*S. pennellii*). (**D**) Heatmap representing the expression profiles of the differentially expressed known miRNAs. (**E**) Heatmap representing the expression profiles of the differentially expressed novel miRNAs.

**Figure 2 ijms-22-12238-f002:**
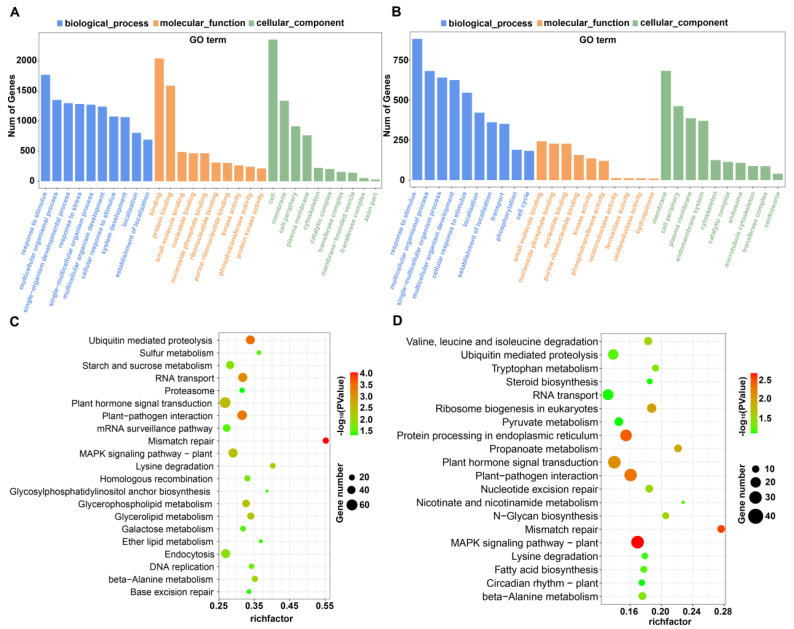
GO and KEGG pathway enrichment analysis results of salt-responsive miRNA target genes. (**A**) GO enrichment results for target genes of DE-miRNAs in M82. (**B**) GO enrichment results for target genes of DE-miRNAs in *Solanum pennellii* (*S. pennellii*). (**C**) KEGG enrichment results for target genes of DE-miRNAs in M82. (**D**) KEGG enrichment results for target genes of DE-miRNAs in *S. pennellii*.

**Figure 3 ijms-22-12238-f003:**
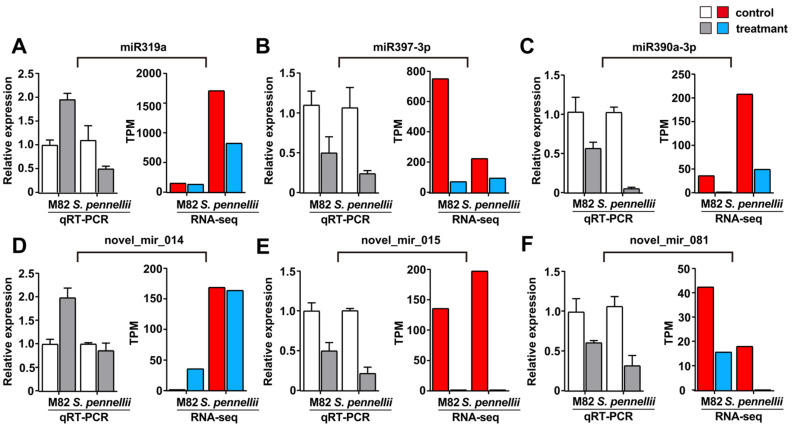
qRT-PCR analysis of the expression patterns of selected DE-miRNAs (three known and three novel miRNAs). (**A**–**F**): The qRT-PCR results are represented by the four columns on the left of each graph, and the expression levels of miRNAs are represented by the four columns on the right. The error bars represent the standard error of replicates. The white and red bars represent the control samples, and the gray and blue bars represent the samples that were treated by salt stress.

**Figure 4 ijms-22-12238-f004:**
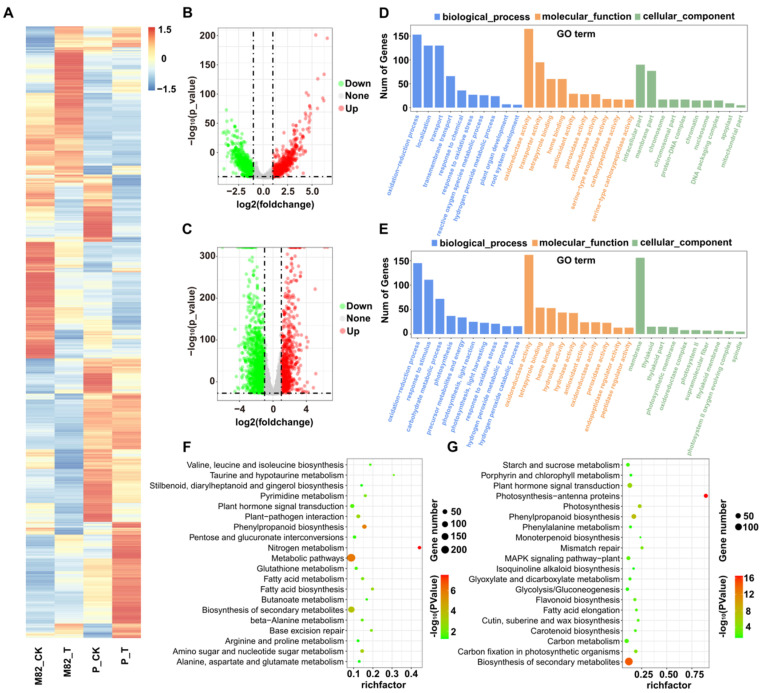
Identification of salt-responsive mRNAs in M82 and *Solanum pennellii* (*S. pennellii*). (**A**) The heatmap represents the expression patterns of all mRNAs. (**B**) The volcano plot represents the DE-mRNAs in M82, and the red and green circles represent the up-regulated and down-regulated mRNAs, respectively. (**C**) The volcano plot represents the DE-mRNAs in *S. pennellii*, and the red and green circles represent the up-regulated and down-regulated mRNAs, respectively. (**D**,**E**) GO enrichment analysis results for DE-mRNAs in M82 (**D**) and *S. pennellii* (**E**). (**F**,**G**) KEGG pathway analysis of DE-mRNAs in M82 (**F**) and *S. pennellii* (**G**).

**Figure 5 ijms-22-12238-f005:**
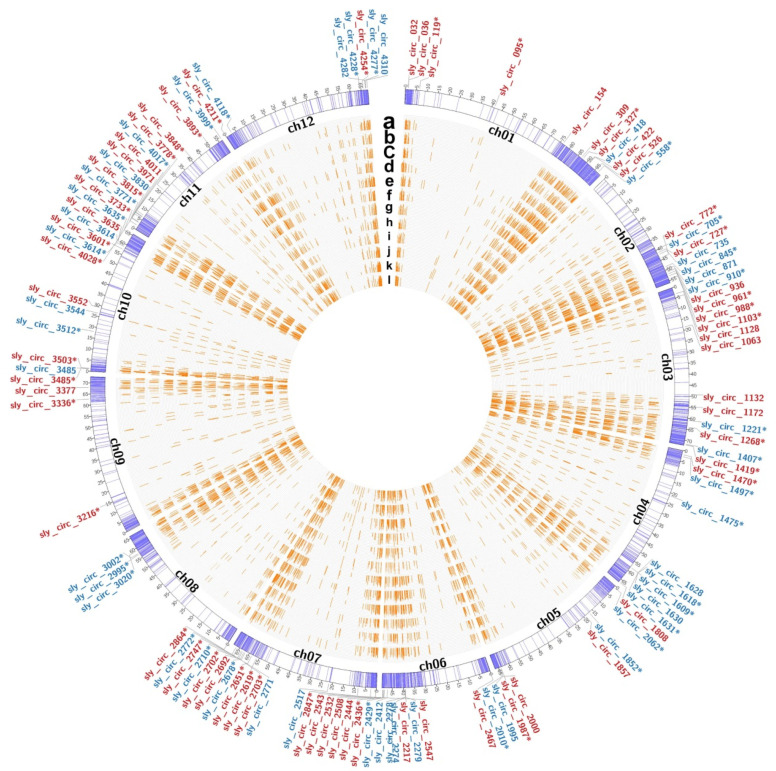
The circos plot of circRNAs shows the distribution of circRNAs in different chromosomes. From outside to inside, the circos plot is divided into three parts. The outermost circle contains the up-regulated (red) and down-regulated (blue) circRNAs. The names of circRNAs with or without asterisks indicate that the circRNAs were from M82 or *Solanum pennellii* (*S. pennellii*), respectively. The middle circle represents the 12 chromosomes of the tomato, and the purple lines represent the locations of the circRNAs identified in this study. The innermost part represents the expression levels of all circRNAs in each sample. From a to l, the 12 samples M82-CK1, M82-CK2, M82-CK3, M82-T1, M82-T2, M82-T3, P-CK1, P-CK2, P-CK3, P-T1, P-T2, and P-T3 are represented sequentially.

**Figure 6 ijms-22-12238-f006:**
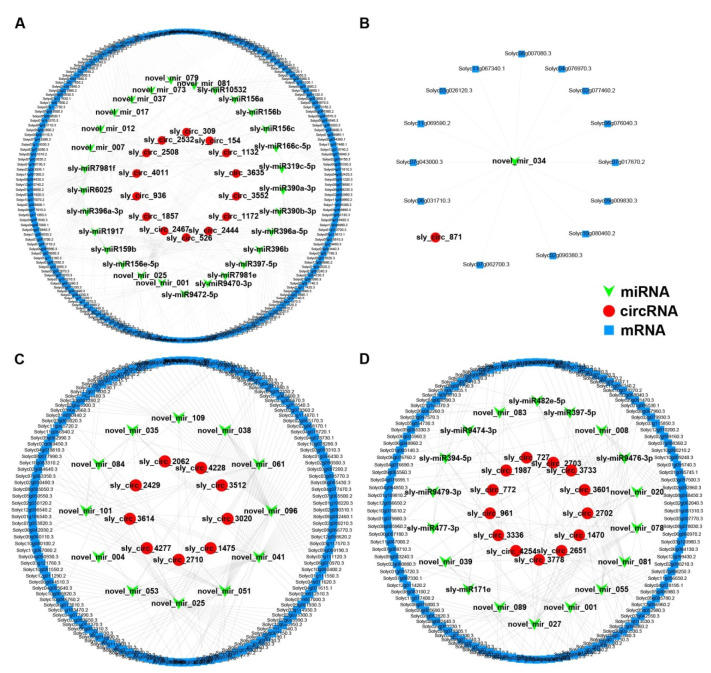
The circle plots of ceRNA (circRNA–miRNA–mRNA) networks. (**A**,**B**) The salt-responsive ceRNA network for M82, which was constructed with circRNA–miRNA and miRNA–mRNA pairs. The circRNAs, miRNAs, and mRNAs in (**A**) are up-regulated, down-regulated, and up-regulated, respectively. The expression patterns are the opposite in (**B**). (**C**,**D**) The salt-responsive ceRNA network for *Solanum pennellii* (*S. pennellii*). The circRNAs, miRNAs, and mRNAs in (**C**) are up-regulated, down-regulated, and up-regulated, respectively. The expression patterns are the opposite in (**D**).

**Figure 7 ijms-22-12238-f007:**
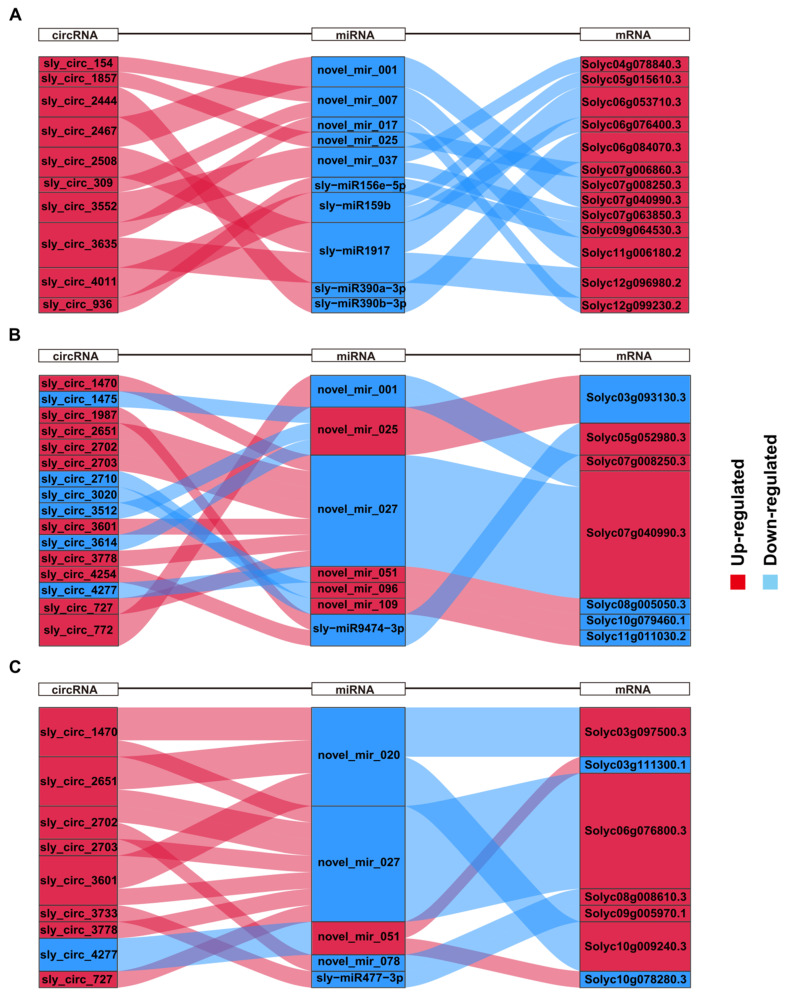
The Sankey plots show the ceRNA networks of three pathways. Red and blue represent up-regulation and down-regulation, respectively. (**A**) The ceRNA network of the plant hormone signal transduction pathway in M82. (**B**) The ceRNA network of the plant hormone signal transduction pathway in *Solanum pennellii* (*S. pennellii*). (**C**) The ceRNA network of the suberin biosynthetic process was enriched only in *S. pennellii*.

**Table 1 ijms-22-12238-t001:** Statistical analysis of small RNA sequencing data for eight samples.

Samples ^1^	Raw Reads	Clean Reads	Length (18–30 nt)	Q20 (%)	Q30 (%)	GC (%)
MCK1	19,164,692	18,960,794	16,466,635	99.38	98.35	51
MCK2	18,813,751	18,678,235	13,675,379	99.59	98.72	52
PCK1	18,479,610	18,396,518	13,819,122	99.56	98.64	55
PCK2	20,555,858	20,428,981	17,783,906	99.59	98.70	54
MT1	21,729,061	21,590,130	15,863,323	99.55	98.62	50
MT2	26,172,258	25,926,977	21,979,146	99.50	98.52	49
PT1	19,849,787	19,700,143	15,616,279	99.57	98.65	52
PT2	19,518,318	19,345,616	13,681,724	99.58	98.69	51

^1^ MCK1/2 and PCK1/2 represent two replicates of the control groups of M82 and *Solanum pennellii* (*S. pennellii*), respectively; MT1/2 and PT1/2 represent two replicates of the salt-treated groups of M82 and *S. pennellii*, respectively.

**Table 2 ijms-22-12238-t002:** Summary of RNA-seq data.

Samples ^1^	Read Number	Base Number	GC Content (%)	Q30 (%)
M82-CK1	136,735,222	10,287,608,907	45	89.42
M82-CK2	122,505,086	9,237,784,418	45	87.07
M82-CK3	127,746,338	9,622,066,421	45	88.25
P-CK1	123,116,402	9,272,390,810	44	89.71
P-CK2	132,255,878	9,958,938,470	43	88.65
P-CK3	128,380,664	9,668,085,519	43	89.25
M82-T1	151,887,684	11,430,890,506	45	89.05
M82-T2	147,518,380	11,100,219,513	47	88.41
M82-T3	150,787,186	11,347,268,749	46	88.80
P-T1	110,834,866	8,292,641,368	43	89.71
P-T2	160,240,518	11,989,101,650	43	86.94
P-T3	148,489,644	11,109,924,975	43	88.08

^1^ M82-CK1/2/3 and P-CK 1/2/3 represent the three replicates of the control groups for M82 and *Solanum pennellii* (*S. pennellii*), respectively; M82-T1/2/3 and P-T1/2/3 represent the three replicates for the salt-treated groups of M82 and *S. pennellii*, respectively.

## Data Availability

The raw sequence data reported in this paper have been deposited in the Genome Sequence Archive (Genomics, Proteomics, and Bioinformatics 2021) in the National Genomics Data Center (Nucleic Acids Res 2021), China National Center for Bioinformation/Beijing Institute of Genomics, Chinese Academy of Sciences, under accession numbers CRA004288 and CRA004289 and are publicly accessible at https://ngdc.cncb.ac.cn/gsa.

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
