# Peer review of "Genome-Wide Characterization of Salt-Responsive miRNAs, circRNAs and Associated ceRNA Networks in Tomatoes"

_ijms, 2021, doi:10.3390/ijms222212238_

Round 1
Reviewer 1 Report
The article “Genome-Wide Characterization of Salt-Responsive MiRNAs, CircRNAs and Associated CeRNA Networks in Tomatoes” was investigated by Wang et al. to aim to provide the mechanism of salt tolerance in response of tomato plants to salt stress. The authors made use of small RNA sequencing, transcriptome analysis to dissect salt-responsive miRNAs, mRNAs and circRNAs from root tissue in tomato plants of M82 and Solanum Pennellii in response to salt stimuli. In addition, the authors illustrated several salt-responsive ceRNA networks, thereby exhibiting that circRNAs presumably function as miRNA sponges to control target mRNA of miRNA. The paper seems to be clear and well-written. The authors provided sufficient data and has well-organized their results and notions. Also, the introduction and the discussion is adequate with their hypothesis.
Author Response
Dear Reviewer:
Thank you for your comments concerning our manuscript entitled “Genome-Wide Characterization of Salt-Responsive MiRNAs, CircRNAs and Associated CeRNA Networks in Tomatoes” (ID: ijms-1445065). Your comments are important guiding significance to our researches. Special thanks to you for your good comments.
Responds to your comments:
Point 1: The article “Genome-Wide Characterization of Salt-Responsive MiRNAs, CircRNAs and Associated CeRNA Networks in Tomatoes” was investigated by Wang et al. to aim to provide the mechanism of salt tolerance in response of tomato plants to salt stress. The authors made use of small RNA sequencing, transcriptome analysis to dissect salt-responsive miRNAs, mRNAs and circRNAs from root tissue in tomato plants of M82 and Solanum Pennellii in response to salt stimuli. In addition, the authors illustrated several salt-responsive ceRNA networks, thereby exhibiting that circRNAs presumably function as miRNA sponges to control target mRNA of miRNA. The paper seems to be clear and well-written. The authors provided sufficient data and has well-organized their results and notions. Also, the introduction and the discussion is adequate with their hypothesis.
Response 1: On behalf of my co-authors, we would like to express our great appreciation to you.
Reviewer 2 Report
Authors described the results of a genome-wide characterization of salt-responsive miRNAs, circRNAs and associated ceRNA networks in tomatoes. They used efficient methods to elucidate the importance of these types of RNAs in tomatoes after salt stress. Results added enormous data to our knowledge about the responses of salt stress of two different tomatoes, the cultivated tomato M82 and wild tomato Solanum pennellii (S. pennellii).
I recommend this manuscript to be accepted for publication after English revision.
Author Response
Dear Reviewer:
Thank you for your comments concerning our manuscript entitled “Genome-Wide Characterization of Salt-Responsive MiRNAs, CircRNAs and Associated CeRNA Networks in Tomatoes” (ID: ijms-1445065). Those comments are all valuable and very helpful for revising and improving our paper, as well as the important guiding significance to our researches. We have studied comments carefully and have made corrections which we hope meet with approval. Revised portions are marked in yellow, pink and grey in the revised manuscript. The main corrections in the paper and the responds to the reviewer's comments are as flowing:
Point 1: Authors described the results of a genome-wide characterization of salt-responsive miRNAs, circRNAs and associated ceRNA networks in tomatoes. They used efficient methods to elucidate the importance of these types of RNAs in tomatoes after salt stress. Results added enormous data to our knowledge about the responses of salt stress of two different tomatoes, the cultivated tomato M82 and wild tomato Solanum pennellii (S. pennellii). I recommend this manuscript to be accepted for publication after English revision. 

Response 1: Specific changes to the revised manuscript that based on your comments are described below:
- We have italicized all the gene names throughout the manuscript, the italics are highlighted with a yellow background.
- On page 1, the following sentence “During the course of evolution, many plants have developed several salt tolerance mechanisms that either exclude salts from cells or tolerate the presence of salt in cells [6]. In the process of plants being exposed to salt stress, like photosynthesis, protein synthesis, energy, and lipid metabolism can be affected [7]” is changed to “In the course of evolution, many plants have evolved several salt tolerance mechanisms for adaptation to an alkaline soil environment [6]. The major plant physiological and metabolic pathways of plants, such as photosynthesis, protein synthesis, energy, and lipid metabolism can be affected under salt stress [7]”
- On page 1, the following words “And plants form a series of mechanisms” are changed to “And plants evolved a series of mechanisms”
- On page 1, we replace "," by a “and” for the following words “ion transport, antioxidants”
- On page 1, the following sentence “such as Solanum pennellii ( pennellii) [9], Solanum peruvianum [10], Solanum Cheesmanii [10], and Solanum chilense [11]” is changed to “such as Solanum pennellii (S. pennellii) [9], S. peruvianum [10], S. Cheesmanii [10], and S. chilense [11]”
- On page 2, the following sentence “Presently, some certain signs of progress of phytohormone signaling and specific transcription factors (TFs) had been achieved and revealed the importance of them under salt stress conditions, including” is changed to “Presently, some studies on phytohormone signaling and specific transcription factors (TFs) had revealed their important roles under salt stress conditions, including”
- On page 2, the following words “Previous researches” are changed to “Previous investigations”
- On page 2, the following words “However, no studies had posed the relation” are changed to “However, no studies had unveiled the relation”
- On page 2, we add a “the” before “shoot in Arabidopsis [39] ”
- On page 2, the following sentence “promote development of lateral root through transcriptional factor BES1” is changed to “promote the development of lateral roots through transcriptional factor BES1”
- On page 2, the following words “become increasingly important” are changed to “become increasingly recognized”
- On page 2, we replace "," by a “and” for the following words “drought [48-53], chilling stress [54-58]”
- On page 3, we add a “be” before “significantly induced in roots”
- On page 3, we change the wrong spelling word “IncRNAs” to “lncRNAs”
- On page 6, the following words “To further investigated” are changed to “To further investigate”
- On page 6, the following words “four above progresses” are changed to “four above progress”
- On page 7, “suggest that the results of Illumina sequencing were reliable” is changed to “suggesting that the results of Illumina sequencing were reliable”
- On page 8, “After remove the primers” is changed to “After removing the primers”
- On page 11, we add a “the” between “While” and “response to stimulus”
- On page 17, we change “homologues” to “homologs”
- On page 21, we change “then the concentration of libraries was” to “then the concentration of libraries were”
We tried our best to improve the manuscript and made some changes in the manuscript. These changes will not influence the content and framework of the paper.
We appreciate your warm work earnestly and hope that the corrections will meet with approval.
Once again, thank you very much for your comments and suggestions.
Reviewer 3 Report
The scientific content of this manuscript is acceptable. The main problem is its English writing.
Use of English terms needs to be more precise and specific. For example, “sequencing” should be “DNA sequencing” in line 29. In line 11, cultivation of salt-tolerant crops is an effective way to “sustain” (rarely can “improve”) crop yield. Line 34 - And plants “evolved” a series of mechanisms to protect themselves from…. Line 54 – Previous “studies” or “investigations” is better than “researches.” Line 70 - no studies had “unveiled” the relation between CDPK16 and salt stress. Lines 82 to 83 - The biological significance of ncRNAs in abiotic stress had become increasingly “recognized” in recent years.
Structure of English sentences should be simple and clear to make them easy to understand. As examples, the sentences on lines 32 to 34 and 41 to 44 are difficult to read or to comprehend the meaning when the subject and the verb are not easily found.
Regarding the nomenclature of species, the genus name must be fully spelled out only when first-time mentioned. Therefore, lines 38 to 39 should be written as “.. such as Solanum pennellii [9], S. peruvianum [10], S. cheesmanii [10], and S. chilense [11].”
When listing a series of things, always use the “and” for the last one. Line 36 – “and” antioxidants; line 89 – “and” chilling stress; etc.
Lines 46 to 47: Presently, some studies on phytohormone signaling and specific transcription factors (TFs) had revealed their important roles under salt stress conditions, including…….
There are some typos. Line 105 - long non-coding RNAs (lncRNAs), it is lower case of L, not capital i.
Author Response
Dear Reviewer:
Thank you for your comments concerning our manuscript entitled “Genome-Wide Characterization of Salt-Responsive MiRNAs, CircRNAs and Associated CeRNA Networks in Tomatoes” (ID: ijms-1445065). Those comments are all valuable and very helpful for revising and improving our paper, as well as the important guiding significance to our researches. We have studied comments carefully and have made corrections which we hope meet with approval. Revised portions are marked in yellow, pink and grey in the revised manuscript. The main corrections in the paper and the responds to the reviewer's comments are as flowing:
Point 1: The scientific content of this manuscript is acceptable. The main problem is its English writing. Use of English terms needs to be more precise and specific.
Response 1: Thank you very much for your appreciation of our work. We also thank your constructive feedback on our English writing problem. We have revised the manuscript according to the suggestions of yours one by one. The specific changes were listed below:
- Point 2: For example, “sequencing” should be “DNAsequencing” in line 29.
Response 2: We have changed “sequencing” to “DNA sequencing”
- Point 3: In line 11, cultivation of salt-tolerant crops is an effective way to “sustain” (rarely can “improve”) crop yield.
Response 3: We have changed “improve” to “sustain”
- Point 4: Line 34 - And plants “evolved” a series of mechanisms to protect themselves from….
Response 4: We have changed “And plants form a series of mechanisms” to “And plants evolved a series of mechanisms”
- Point 5: Line 54 – Previous “studies” or“investigations” is better than “researches.”.
Response 5: We have changed “Previous researches” to “Previous investigations”
- Point 6: Line 70 - no studies had “unveiled” the relation between CDPK16 and salt stress.
Response 6: We have changed “no studies had posed the relation” to “no studies had unveiled the relation”
- Point 7: Lines 82 to 83 - The biological significance of ncRNAs in abiotic stress had become increasingly “recognized” in recent years.
Response 7: We have changed “The biological significance of ncRNAs in abiotic stress had become increasingly important in recent years” to “The biological significance of ncRNAs in abiotic stress had become increasingly recognized in recent years”
- Point 8: Structure of English sentences should be simple and clear to make them easy to understand. As examples, the sentences on lines 32 to 34 and 41 to 44 are difficult to read or to comprehend the meaning when the subject and the verb are not easily found.
Response 8: We have changed the sentences on lines 32 to 34 to “In the course of evolution, many plants have evolved several salt tolerance mechanisms for adaptation to an alkaline soil environment [6]. The major plant physiological and metabolic pathways of plants, such as photosynthesis, protein synthesis, energy, and lipid metabolism can be affected under salt stress [7]”
We have deleted the sentences on lines 41 to 44 “Taking advantage of the convenience of genetic information exchanged between wild tomato and cultivated tomato populations to study the salt tolerance mechanism of wild tomatoes to improve the cultivation of tomatoes is an effective way to cultivate new salt-tolerant cultivars.”
- Point 9: Regarding the nomenclature of species, the genus name must be fully spelled out only when first-time mentioned. Therefore, lines 38 to 39 should be written as “.. such as Solanum pennellii [9], peruvianum [10], S. cheesmanii [10], and S. chilense [11].”
Response 9: We have changed the words on lines 38 to 39 to “such as Solanum pennellii [9], S. peruvianum [10], S. cheesmanii [10], and S. chilense [11].”
- Point 10: When listing a series of things, always use the “and” for the last one. Line 36 – “and” antioxidants; line 89 – “and” chilling stress; etc.
Response 10: We have replaced the “,” with “and” in “ion transport, antioxidants”
- Point 11: Lines 46 to 47: Presently, some studieson phytohormone signaling and specific transcription factors (TFs) had revealed their important roles under salt stress conditions, including….
Response 11: We have changed the sentence “Presently, some certain signs of progress of phytohormone signaling and specific transcription factors (TFs) had been achieved and revealed the importance of them under salt stress conditions” to “Presently, some studies on phytohormone signaling and specific transcription factors (TFs) had revealed their important roles under salt stress conditions, including”
- Point 12: There are some typos. Line 105 - long non-coding RNAs (lncRNAs), it is lower case of L, not capital i.
Response 12: We have changed “IncRNAs” to “lncRNAs”
Some other changes were listed below based on your suggestions:
- We have also italicized all the gene names throughout the manuscript, the italics are highlighted with a yellow background.
- On page 2, we add a “the” before “shoot in Arabidopsis [39] ”
- On page 2, the following sentence “promote development of lateral root through transcriptional factor BES1” is changed to “promote the development of lateral roots through transcriptional factor BES1”
- On page 2, we add a “the” before “shoot in Arabidopsis [39] ”
- On page 2, the following sentence “promote development of lateral root through transcriptional factor BES1” is changed to “promote the development of lateral roots through transcriptional factor BES1”
- On page 3, we add a “be” before “significantly induced in roots”
- On page 6, the following words “To further investigated” are changed to “To further investigate”
- On page 6, the following words “four above progresses” are changed to “four above progress”
- On page 7, “suggest that the results of Illumina sequencing were reliable” is changed to “suggesting that the results of Illumina sequencing were reliable”
- On page 8, “After remove the primers” is changed to “After removing the primers”
- On page 11, we add a “the” between “While” and “response to stimulus”
- On page 17, we change “homologues” to “homologs”
- On page 21, we change “then the concentration of libraries was” to “then the concentration of libraries were”
We tried our best to improve the manuscript and made some changes in the manuscript. These changes will not influence the content and framework of the paper.
We appreciate your warm work earnestly and hope that the corrections will meet with approval.
Once again, thank you very much for your comments and suggestions.